# TransGesture: Autoregressive Gesture Generation with RNN-Transducer

NAOSHI KANEKO, YUNA MITSUBAYASHI, and GENG MU, Aoyama Gakuin University, Japan

This paper presents a gesture generation model based on an RNN-transducer, submitted to the GENEA Challenge 2022. The proposed model consists of three neural networks: Encoder, Prediction Network, and Joint Network, which can be jointly trained in an end-to-end manner. We also introduce new loss functions, namely statistical losses, as the additional term to the standard MSE loss to put motion statistics of generated gestures close to the ground truths'. Finally, we show the subjective evaluation results and discuss the results and takeaways from the challenge.

CCS Concepts: • **Human-centered computing** → **Empirical studies in HCI**; • **Computing methodologies** → **Neural networks**.

Additional Key Words and Phrases: gesture generation, speech audio, neural networks, deep learning

**ACM Reference Format:**
Naoshi Kaneko, Yuna Mitsubayashi, and Geng Mu. 2022. TransGesture: Autoregressive Gesture Generation with RNN-Transducer. In *INTERNATIONAL CONFERENCE ON MULTIMODAL INTERACTION (ICMI '22), November 7–11, 2022, Bengaluru, India.* ACM, New York, NY, USA, 7 pages. https://doi.org/10.1145/3536221.3558061

## 1 INTRODUCTION

Non-verbal information, especially gestures, play an important role in human communication by emphasizing, supporting, or complementing speech [18, 25]. Researchers have discovered that introducing non-verbal expressions into robots or embodied agents has positive effects on human-agent interaction [24, 27]. Thus, enabling such robots/agents to accompany their speech with gestures is important to facilitate natural human-agent interaction.

As the manual implementation of gestures is time- and labor-consuming, automatic gesture generation has attracted the community's attention. Early studies use rule-based methods for the task [5, 17]. Rule-based approaches have the advantage that if enough task- or domain-specific knowledge is available, a system will provide explainable, reasonable gestures. However, collecting such specific knowledge and writing rules take a lot of effort, and the produced gestures have limited variety.

Using data-driven approaches is a promising direction to overcome the limitations of rule-based methods, and it has been the mainstream of gesture generation research nowadays. Researchers have explored several input modalities, including audio, text, speaker identities, and their combinations. In single-modality approaches, using audio as input is a dominant choice due to its good properties, including a strong temporal correlation with gesture motion [7, 8, 12, 15, 20, 23, 26]. Several studies leverage text transcriptions as inputs to generate semantic gestures [1, 4, 32]. Researchers also develop multi-modal methods that can utilize both the acoustic and semantic features [21] as well as speaker identities [3, 31].

From another perspective, a long-time trend in automatic gesture generation is adopting architectures of speech recognition and machine translation. For example, DeepSpeech [14], Seq2Seq [29], and Transformers [30] are used in many gesture generation studies [1, 4, 15, 20, 32]. These models are broadly categorized into *encoder-based models* (e.g.,

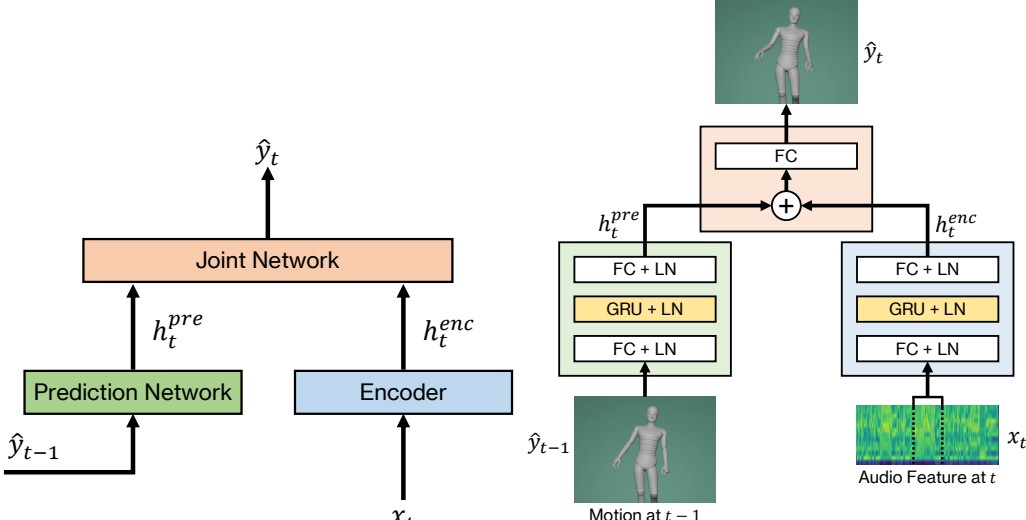

Fig. 1. A schematic diagram of RNN-transducer.

Fig. 2. Architecture of our gesture generation model.

DeepSpeech) and *attention-based encoder-decoder models* (e.g., attention Seq2Seq and Transformers). However, in the speech recognition field, the third type of model, namely *RNN-transducers* [10], has attracted recent attention from the community [13, 16, 28].

An RNN-transducer is composed of three networks, namely Encoder, Prediction Network, and Joint Network, as depicted in Figure 1. Encoder is similar to the ones used in encoder-based models. It takes input signals (e.g., audio or text) and maps them to latent representation. Prediction Network is autoregressive; it takes the transducer's previous output and maps it to another latent representation, which is useful to predict the next output. Finally, Joint Network takes the two latent representations from Encoder and Prediction Network, joins them, and produces outputs. RNN-transducers have several good properties for gesture generation over encoder-based models and attention-based encoder-decoder models. Unlike encoder-based models, RNN-transducers are autoregressive; their successive outputs are assumed to be not independent. With this property, RNN-transducers are supposed to produce smoother and more continuous motions than encoder-based models. On the other hand, attention-based encoder-decoder models require whole input at once to compute attention, whereas RNN-transducers can be streamable. Streamable models are particularly useful for real-time, interactive applications such as social robots and avatar-mediated communications. Despite a wide range of potential applications, most of the previous gesture generation studies did not pay attention to the streamable nature of the models. Note that, other than attention-based encoder-decoder models, bi-directional RNN-based models [15] and the methods using contextual inputs [21] also require features of future timesteps.

To the best of our knowledge, RNN-transducers have not been adopted for gesture generation despite these preferable properties. In this paper, we examine an RNN-transducer-based model for gesture generation. In addition, we introduce statistical loss functions to put the motion statistics of generated gestures close to the ground truths'.

## 2 METHOD

### 2.1 Input and Output Representations

Following the previous studies of audio-driven gesture generation methods, we choose mel-frequency cepstral coefficients (MFCC) as input audio features. First, we downsample the raw audio files from 44.1 kHz to 16 kHz. We extract 13 MFCC features from mel spectrograms of 26 mel filter banks, extracted from the raw audio files at 30 frames per second (FPS) using Torchaudio 0.10.1. We set a window length to 0.025 seconds and set the hop length to the reciprocal of the FPS (i.e., 1/30 seconds).

The output motion representation is joint angles relative to a T-pose, which are parameterized using the exponential map [9] at 30 FPS. Each dimension of the joint angles is normalized by subtracting the mean and dividing by the absolute maximum over the training set, resulting in the range of $[-1, 1]$. For ease of training, we consider the "upper body" (excluding fingers) joint set.

### 2.2 Network Architecture

Our gesture generation model is an RNN-transducer, as shown in Figure 2. The model is composed of three networks, namely Audio Encoder, Prediction Network, and Joint Network. Each network has 256 channels in its hidden layers.

Audio Encoder takes MFCC features at time $t$, namely $x_t$, and processes it through a set of a fully connected layer, a two-layered Gated Recurrent Unit (GRU) [6], and a fully connected layer, resulting in a latent feature vector $h_t^{enc}$. We apply Layer Normalization [2] to each layer.

Prediction Network is autoregressive. It takes the previous output of Joint Network, $\hat{y}_{t-1}$, and produces a latent feature $h_t^{pre}$, which is taken by Joint Network to estimate the output of current time $\hat{y}_t$. We use the same architecture for Prediction Network and Audio Encoder.

Joint Network takes the two networks' outputs, $h_t^{enc}$ and $h_t^{pre}$, to outputs motion of current time $\hat{y}_t$. $h_t^{enc}$ and $h_t^{pre}$ are integrated by element-wise summation followed by a fully connected layer. The output from the last fully connected layer is activated by a hyperbolic tangent (tanh) function to have the value range of $[-1, 1]$.

### 2.3 Loss Function and Optimizer

Our first loss function is the Mean Squared Error (MSE) between the model outputs and ground truth motions in the exponential map representation. The MSE loss is given by

$$L_{mse} = \frac{1}{m} \sum_{i=1}^{m} (\hat{y}_i - y_i)^2, \tag{1}$$

where $m$ is the mini-batch size, $i$ is the index of each sample in the mini-batch, $y$ is the ground truth exponential map, and $\hat{y}$ is the model output. We also compute the MSE loss on velocity [15, 21], which is given by

$$L_{vel} = \frac{1}{m} \sum_{i=1}^{m} (\Delta \hat{y}_i - \Delta y_i)^2, \tag{2}$$

where $\Delta$ indicates the finite differences between adjacent frames.

While MSE is the standard loss function used in gesture generation, we observe that our network outputs tend to collapse to the mean pose. One way to prevent such collapse is adding adversarial discriminator loss, e.g., as done in [8]. However, we empirically found that such adversarial losses tend to unstabilize the training and produce "messy", inhuman-like gestures. Thus, we introduce new loss functions, namely statistical losses. The statistical losses ensure

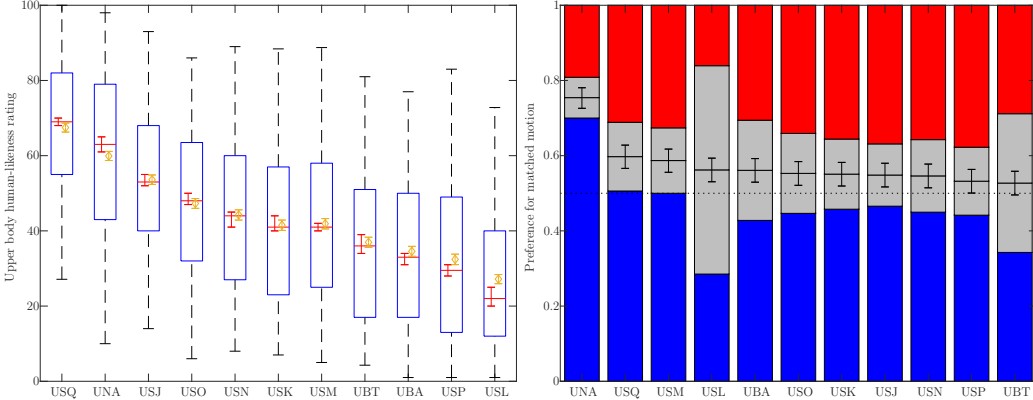

Fig. 3. Human-likeness evaluation result.    Fig. 4. Appropriateness evaluation result.

that the statistical characteristics of the generated and ground truth gestures are close. Specifically, our system uses the mean and variance of the gestures, i.e.,

$$L_{mean} = \frac{1}{m} \sum_{i=1}^{m} (Mean(\hat{y}_i) - Mean(y_i))^2, \tag{3}$$

$$L_{var} = \frac{1}{m} \sum_{i=1}^{m} (Var(\hat{y}_i) - Var(y_i))^2. \tag{4}$$

The final loss function is the weighted sum of the above four terms,

$$L = L_{mse} + \lambda_1 L_{vel} + \lambda_2 L_{mean} + \lambda_3 L_{var}, \tag{5}$$

where we empirically set to $\lambda_1 = 1$, $\lambda_2 = 2$, and $\lambda_3 = 5$. We trained the model by Adam optimizer [19] with a learning rate of 0.001, and a mini-batch size of 32. We ran the model training for 10 epochs, where the improvement in validation loss had stopped. We tuned the hyperparameters by manually watching the generated gestures in the validation set.

## 3 EXPERIMENT AND RESULT

**Dataset.** The challenge dataset is Talking With Hands 16.2M [22]. We used the official training, validation, and test splits provided by the challenge organizers. Our system was trained on the training split only. We did not perform sample selection, i.e., we used all the samples in the training set. For the details of the challenge dataset, please refer to the challenge paper [33].

**Implementation.** We implemented our system using Python 3.9.7, PyTorch 1.10.1, and Torchaudio 0.10.1. We divided each training sample into smaller chunks of 256 frames (about 8.53 seconds in 30 FPS) to make the memory footprint reasonable. For BVH file processing and motion parameterization, we used PyMO library provided by the challenge organizers. Before converting the generated exponential maps into BVH files, we applied a Savitzky-Golay filter, as the audio-only baseline system suggested.

**Evaluation.** As described in the challenge paper [33], this year's challenge evaluated the submitted gestures from two aspects: human-likeness and appropriateness. The label indicating our challenge entry is "USL".

Table 1. Motion statistics with or without the statistical losses. The numbers in parentheses are absolute differences against the ground truth values.

| Condition | Mean | Standard Deviation |
|---|---|---|
| GT | 0.7746 | 0.0829 |
| w/o statistical losses | 0.7723 (0.0023) | 0.0124 (0.0705) |
| w/ statistical losses | **0.7762 (0.0016)** | **0.0320 (0.0509)** |

### 3.1 Human-Likeness Evaluation

In this evaluation, participants were asked to watch the *silent* videos of gesture motions and evaluate the human-likeness of the gesture motions with 0-100 score. The evaluation result is presented in Figure 3. Our entry, USL, attained the lowest median score among the participating systems. There are several possible reasons for the low score. First, our entry did not contain finger movements as described in Section 2.1.[1] Second, in some of the evaluation videos, our entry contained no clear gesticulation, i.e., only body sway was performed. Third, we observed that some of our generated gestures were somewhat too symmetric; both arms did similar movements, with similar height and speed. Thus, the evaluation participants might have given low scores to such videos.

### 3.2 Appropriateness Evaluation

In the appropriateness evaluation, participants were asked to watch videos of gesture motions *with speech audio*. The participants watched two videos side-by-side: one generated from the matched speech and the other one generated from the mismatched speech, and selected which one was more appropriate (or they were equal) for the given speech. The result is shown in Figure 4, where the blue bar, grey bar, and red bar represent the ratio of responses preferring the matched, tied, and mismatched motions, respectively. Our entry USL has the largest grey bar among all the entries, implying its gesticulation was somewhat obscured.

### 3.3 Effect of the Statistical Losses

We additionally evaluated how the proposed statistical losses affected the motion statistics of the generated gestures. Specifically, we compared the mean and standard deviation values between the generated and ground truth gestures, represented as exponential maps. We used the validation splits for the comparison. Table 1 shows that the statistical losses slightly improve the motion characteristics of the generated gestures, pushing toward the ground truth values. However, the standard deviation of the generated gestures was still significantly smaller than the ground truths'. The result implies that the generated gestures had a limited motion variety, which may support the results of the subjective evaluations.

## 4 CONCLUSIONS AND TAKEAWAYS

This paper has presented our entry for the GENEA challenge 2022. The proposed model is based on an RNN-transducer consisting of three neural networks: Encoder, Prediction Network, and Joint Network. In addition, we incorporate statistical losses to let generated gestures have close motion statistics with the ground truths'.

Honestly speaking, we had had difficulty in generating diverse, meaningful gestures from the challenge dataset. We had spent too much time implementing and debugging the prototype system and had not had much time to tune up

---

[1]The text-only baseline (labelled UBT) also did not contain the finger motion.

and improve it. Since the challenge dataset (Talking With Hands 16.2M) is in a two-person conversational setting, the dataset contains some amount of silence ("listening") frames. Generally, when speech stops, gesticulation also stops [11]; since we divided the training samples into chunks, some chunks barely contained gesticulated frames. Exploring a filtering strategy for such chunks (e.g., thresholding by the number of spoken words) would be useful. Also, we found that normalization on the output representation affected the gesture quality; our system could not learn gesticulation from unnormalized exponential maps. Since the text- and audio-only baseline employed different preprocessing steps and output representations (joint coordinates + rotation matrix vs. exponential maps), we took some time to compare and analyze the baseline codes. We have learned and experienced much from this challenge.

## ACKNOWLEDGEMENT

This work was partially supported by JSPS KAKENHI Grant Number JP21K12160.

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
