# OpenReview forum: "TransGesture: Autoregressive Gesture Generation with RNN-Transducer"
_ACM.org/ICMI/2022/Workshop/GENEA — GENEA Challenge & Workshop 2022 Mainproceeding_

### Official Review · Reviewer_MC6E · 2022-07-30
**No need for correction**

**Rating:** 8
**Confidence:** 3

**Review:**

Overall, the paper does require any correction. The provided approach is clear and observations of predicted motion behavior are also presented. However, there are some aspects to discuss.

In Kucherenko et al. 2020 authors provided an approach reminiscent of your. It also used auto-regressive input to predict the subsequent pose. Furthermore, authors faced the same problem with collapsing to the mean pose. They solved it  with a special teacher-forcing strategy. It seems that only adding losses on mean and variance is not enough to solve this problem. Your model could be just trained to continue motion and do not extract information from audio. Maybe changing the training strategy will fix it.

---

### Official Review · Reviewer_Gozb · 2022-07-31
**An interesting approach, but questions remain on the quality of the network's convergence**

**Rating:** 5
**Confidence:** 5

**Review:**

## Strengths
- The proposed approach is sound, well-motivated, and well-explained.

- The ability to predict gestures with partial audio inputs (aka the "streaming" approach) is particularly and can be potentially useful for real-time applications down the line.

## Weaknesses
- Many missing references:

[A] Taras Kucherenko, Patrik Jonell, Sanne van Waveren, Gustav Eje Henter, Simon Alexandersson, Iolanda Leite, and Hedvig Kjellström. "Gesticulator: A framework for semantically-aware speech-driven gesture generation." In Proceedings of the 2020 International Conference on Multimodal Interaction, pp. 242-250. 2020.

[B] Uttaran Bhattacharya, Nicholas Rewkowski, Abhishek Banerjee, Pooja Guhan, Aniket Bera, and Dinesh Manocha. "Text2gestures: A transformer-based network for generating emotive body gestures for virtual agents." In 2021 IEEE Virtual Reality and 3D User Interfaces (VR), pp. 1-10. IEEE, 2021.

[C] Uttaran Bhattacharya, Elizabeth Childs, Nicholas Rewkowski, and Dinesh Manocha. "Speech2affectivegestures: Synthesizing co-speech gestures with generative adversarial affective expression learning." In Proceedings of the 29th ACM International Conference on Multimedia, pp. 2027-2036. 2021.

[D] Ylva Ferstl, Michael Neff, and Rachel McDonnell. 2021. ExpressGesture: Expressive gesture generation from speech through database matching. Computer Animation and Virtual Worlds 32 (6 2021), e2016. Issue 3-4. https://doi.org/10.1002/CAV.2016

[E] Jing Li, Di Kang, Wenjie Pei, Xuefei Zhe, Ying Zhang, Zhenyu He, and Linchao Bao. 2021. Audio2Gestures: Generating Diverse Gestures from Speech Audio with Conditional Variational Autoencoders. 2021 IEEE/CVF International Conference on Computer Vision (ICCV) (10 2021), 11273–11282. https://doi.org/10.1109/ICCV48922.2021.01110

[F] Shenhan Qian, Zhi Tu, Yihao Zhi, Wen Liu, and Shenghua Gao. 2021. Speech Drives Templates: Co-Speech Gesture Synthesis with Learned Templates. 2021 IEEE/CVF International Conference on Computer Vision (ICCV) (10 2021), 11057–11066. https://doi.org/10.1109/ICCV48922.2021.01089

[G] Ikhsanul Habibie, Mohamed Elgharib, Kripashindu Sarkar, Ahsan Abdullah, Simbarashe Nyatsanga, Michael Neff, and Christian Theobalt. 2022. A Motion Matching-based Framework for Controllable Gesture Synthesis from Speech. In SIGGRAPH ’22 Conference Proceedings.

- The approach of using statistical losses is not fully clear to me. Based on Eqs. 2 and 3, the statistical losses depend on the mini-batch size $m$. Did the authors perform experiments to observe any potential correlations between the stability of their statistical losses and the value of $m$?

- 10 epochs seems to be an unusually low number of epochs for the validation loss to saturate at. Did the authors use any regularization loss on the network parameters or incorporate any weight decay in their optimizer?

- "We tuned the hyperparameters [...] manually": what range of hyperparameters did the authors experiment with?

## Minor Comments
- Sec 2.3, para 2: "as done in []." The reference number is missing here.

## Overall
While the paper presents a decent approach to synthesizing co-speech gestures, the design of the statistical losses (especially with the dependence on the mini-batch size) and the very early convergence (within 10 epochs) indicates a potentially bad convergence of the network to a sub-optimal local minimum. The comparatively poor performance on the evaluation benchmarks potentially provides further evidence of the said bad convergence.

---

### Official Review · Reviewer_KsvN · 2022-08-06
**A novel attempt for gesture generation based on an RNN-Transducer**

**Rating:** 6
**Confidence:** 3

**Review:**

This paper proposes a novel model based on RNN-Transducer for audio-to-gesture generation with preliminary experimental results showing the feasibility of the proposed model.

The technical descriptions are simple but clear enough to understand the proposed approach. The performance of the approach is not satisfactory but it is a valuable contribution to the workshop in two points: 1) a novel model using RNN-T is proposed, 2) some comments on the challenge dataset is made which could contribute to the improvement of the challenge.

Some comments for improving the paper:
1) The paper needs editorial improvements. A thorough proof-reading is recommended. ex) a missing citation number in the 2nd paragraph of section 2.3.
2) It could have been more informational if the benefit of using statistical loss terms had been explained with quantitative or qualitative evaluations.
3) In section 4, it is mentioned that the challenge "dataset contains a non-negligible amount of non-gesticulated frames." Presenting more details on the issue in a separate section would be an invaluable contribution to the challenge and the workshop.
4) As mentioned by the authors in section 1, RNN-T model can process input in a streaming fashion. The authors need to elaborate some more to explain why this property of RNN-T is beneficial in gesture generation.

---

### Decision · Program_Chairs · 2022-08-11

**Decision:**

Accept (Main proceeding)

**Comment:**

A majority of reviewers recommended accepting this paper. Reviewers especially appreciated the streaming nature of the proposed approach. As for weaknesses, several reviewers commented on the motivation for the statistical losses, the editing/proofreading, and the referencing/discussion of prior work.

Based on the reviews, the chairs decided to accept this paper into the main ACM ICMI proceedings.

For the final (camera-ready) submission, the chairs recommend the following changes:

1) Expand on the motivation regarding the advantages and drawbacks of streamable models. The fact that non-streamable models are less suitable for interactive applications is one thing to highlight.

2) Add some motivation and support for the specific statistical losses chosen. Quantitative results would be ideal, if the authors have them, but there is likely not enough time to compute new results or carry out new studies before the camera-ready deadline. Since the model converges in only 10 epochs, perhaps it may be possible to (after the camera-ready deadline) upload videos of models that ablate these various loss terms, to a location where readers of the paper reader can see these videos and get an idea of the effects of the different terms?

3) Consider incorporating the references listed by reviewer Gozb. Comment on the similarities to, and differences from, reference [A] in particular, since this reference was also pointed out by reviewer MC6E as well.

4) Be a little more explicit about how the presence of “non-gesticulated frames” might have affected the output. As in, in what way would a filtering strategy be likely to improve the output?

5) Proof-read and edit the submission as suggested by the reviewers.

Since a short paper can be up to 7 pages (single-column format, not counting references), there should be plenty of space to make useful additions.